# Computational prediction of ω-transaminase selectivity by deep learning analysis of molecular dynamics trajectories

Deep learning; enantioselectivity predictions; molecular dynamics trajectories; ω-transaminases

**Author for correspondence:**
*Siewert J. Marrink,
E-mail: s.j.marrink@rug.nl

Carlos Ramírez-Palacios 🄳 and Siewert J. Marrink* 🄳

Molecular Dynamics, Groningen Biomolecular Sciences and Biotechnology Institute (GBB), University of Groningen, Nijenborgh 7, 9747 AG Groningen, The Netherlands

## Abstract

We previously presented a computational protocol to predict the enzymatic (enantio)selectivity of an ω-transaminase towards a set of ligands (Ramírez-Palacios *et al.* (2021) *Journal of Chemical Information and Modeling* 61(11), 5569–5580) by counting the number of binding poses present in molecular dynamics (MD) simulations that met a defined set of geometric criteria. The geometric criteria consisted of a hand-crafted set of distances, angles and dihedrals deemed to be important for the enzymatic reaction to take place. In this work, the MD trajectories are reanalysed using a deep-learning approach to predict the enantiopreference of the enzyme without the need for hand-crafted criteria. We show that a convolutional neural network is capable of classifying the trajectories as belonging to the 'reactive' or 'non-reactive' enantiomer (binary classification) with a good accuracy (>0.90). The new method reduces the computational cost of the methodology, because it does not necessitate the sampling approach from the previous work. We also show that analysing how neural networks reach specific decisions can aid hand-crafted approaches (e.g. definition of near-attack conformations, or binding poses).

## Introduction

The information obtained from molecular dynamics (MD) simulations of protein–ligand complexes can be leveraged to predict enzymatic activity. The most common strategy is to use the MD trajectory to compute the binding energy of the complex, which is expected to be correlated with the enzymatic activity (Limongelli, 2020). The main challenge of calculating binding energies from simulations is that enough conformational space needs to be explored, making the approach expensive (Li and Gilson, 2018). Another strategy is to directly analyse how the ligand interacts with the enzyme during the simulation. The behaviour of the complex of interest during the simulation can hint to how well the enzyme will be able to accommodate the ligand, and hence catalyse the reaction (Bruice, 2002; Voss *et al.,* 2018; Ramírez-Palacios *et al.,* 2021). However, identification of the set of geometric criteria that can distinguish good from bad ligands or enzymes is a laborious task.

Deep learning (DL) can help in the identification of interesting molecular events taking place during an MD simulation (Berishvili *et al.,* 2019; Terao, 2020; Wang *et al.,* 2020). The main advantage of DL is the reduced manual intervention needed to perform analysis of data, often yielding better accuracies than traditional methods (i.e. hard-coded algorithms). The high-dimensionality of MD trajectories complicates any analysis, and DL can help alleviate this factor by, for example, meaningfully projecting the high-dimensional (high-$d$) trajectory into a low-$d$ space to facilitate visualisation and analysis (Taufer *et al.,* 2020; Frassek *et al.,* 2021; Glielmo *et al.,* 2021). There are several possibilities for projecting simulations into low-$d$ representations, but which one to use will depend on the type of representation used to describe the trajectories. For example, one can view trajectories as a regular-grid of pixels (images), and use the repertory of computer vision machine learning methods to perform the desired task. Or one can view trajectories as time-series data, and use DL to perform time-series forecasting.

The architecture of excellence in DL is the convolutional neural network (CNN; LeCun *et al.,* 1989). CNNs work by sequentially performing convolution operations on subsets of pixels across the entire input image. The convolution operation between an input signal, $f$, and a filter, $g$, can be computed as: $s_t = (f \star g)_t = \sum_{a=-\infty}^{a=\infty} f_{(a)} g_{(a+t)}$, where $s_t$ is the feature map. The filter, $g$, is learned through training. The most popular CNN is 2D-CNN, which works on 2D images, but a simplified version, the 1D-CNN, is more appropriate for trajectories (Jiang and Zavala, 2021). A 3D-CNN has also been proposed (Tran *et al.,* 2015) to be used in molecular representations but it is not as common (Fukuya and Shibuta, 2020). In this work, both 1D-CNN and 2D-CNN are used to analyse the MD trajectories.

Additionally, analysis of MD simulations by DL algorithms can be done through time-series forecasting. Time-series forecasting aims to making predictions about the future values of a series



CAMBRIDGE
UNIVERSITY PRESS

based on its history (e.g. predict tomorrow's temperature given historical weather data of temperature and humidity). Recurrent neural networks (RNNs) are the standard architecture for time-series forecasting (Petneházi, 2019; Hewamalage *et al.,* 2021). A potential pitfall of RNN-based models is that they can suffer from vanishing or exploding gradients if the distance from the input to the output layers becomes too large (Bengio *et al.,* 1994). The most popular RNNs are long short-term memory (LSTM; Hochreiter and Schmidhuber, 1997) and the simpler gated recurrent unit (GRU; Chung *et al.,* 2014). Both LSTM and GRU use gating units to control the flow of information from the distant past to the distant future, preventing degradation of the input signal (Sherstinsky, 2020). A promising alternative for long sequences are attention-based architectures (Vaswani *et al.,* 2017). Autoencoders (AEs) have also been used for time-series forecasting of MD trajectories (e.g. time-lagged AEs), where the aim of the AE is not to reproduce the current but a future simulation frame (Wehmeyer and Noé, 2018). The LSTM architecture was chosen to model the time-series, because of its ability to identify patterns at multiple frequencies (Lange *et al.,* 2020), and molecular events emerge from the combination of molecular motions taking place at different time scales. A neural network trained on a time-series forecasting task may implicitly learn to separate trajectories in classes. Then, projecting the latent space representations of the trajectories into some orthogonal vector in a way that reproduces proximity or passing the embeddings through a dense layer should allow for binary classification. Algorithms for making meaningful projections from high-$d$ to low-$d$ representations are principal component analysis (PCA; Wold *et al.,* 1987), time-independent component analysis (TICA; Molgedey and Schuster, 1994), t-distributed stochastic neighbour embedding (t-SNE; van der Maaten and Hinton, 2008), among others (Das *et al.,* 2006; Ceriotti *et al.,* 2011; Ferguson *et al.,* 2011; Spiwok and Králová, 2011; Tribello *et al.,* 2012; Noé and Clementi,

2015; Chen and Ferguson, 2018; Lemke and Peter, 2019; Spiwok and Kříž, 2020). t-SNE aims to reproduce proximities rather than distances (PCA) or divergences (TICA).

ω-Transaminases are pyridoxal-5′-phosphate (PLP)-dependent enzymes that can catalyse the conversion of chiral amines from more accessible achiral ketones (Cassimjee *et al.,* 2015). ω-Transaminases are desirable catalysts in industry for the production of chiral amines but the substrate scope and selectivity usually need to be fine-tuned to the molecule of interest (Breuer *et al.,* 2004; Kelly *et al.,* 2018). Hence, the development of computational algorithms to predict the selectivity of ω-transaminases can accelerate enzyme design and discovery campaigns. Previously, we presented a framework for predicting the enantiopreference of an ω-transaminase from *Vibrio fluvialis* (*Vf*-TA; Ramírez-Palacios *et al.,* 2021). The approach consisted in counting the number of near-attack conformations (NACs) observed during MD trajectories to quantitatively predict the enantiopreference of the enzyme towards a given ligand (*Fig. 1a*). However, some fine-tuning was needed to find appropriate geometric criteria to define a NAC. Herein, we use a machine learning approach to tackle the same problem (*Fig. 1b*). The MD trajectories are viewed as vectors consisting of descriptors (distances, angles and dihedrals) and labelled as 'reactive' when containing the preferred enantiomer and as 'non-reactive' otherwise. The dataset to train the DL models consists of 100 examples per class and per ligand, summing up to a total of ($100 \times 2 \times 49$) 9,800 examples, split into training (80%) and validation (20%) datasets. Binary classification is achieved by supervised (CNN) or semi-supervised (LSTM) training of a neural network.

The trained CNNs achieved excellent accuracy in the binary classification task. Nonetheless, from a molecular modelling perspective, it is more useful if explanations can also be retrieved from the trained models to identify descriptors that the CNN found to be relevant in achieving the assigned task. Knowledge on the importance of each descriptor can be useful, for example, to refine the set

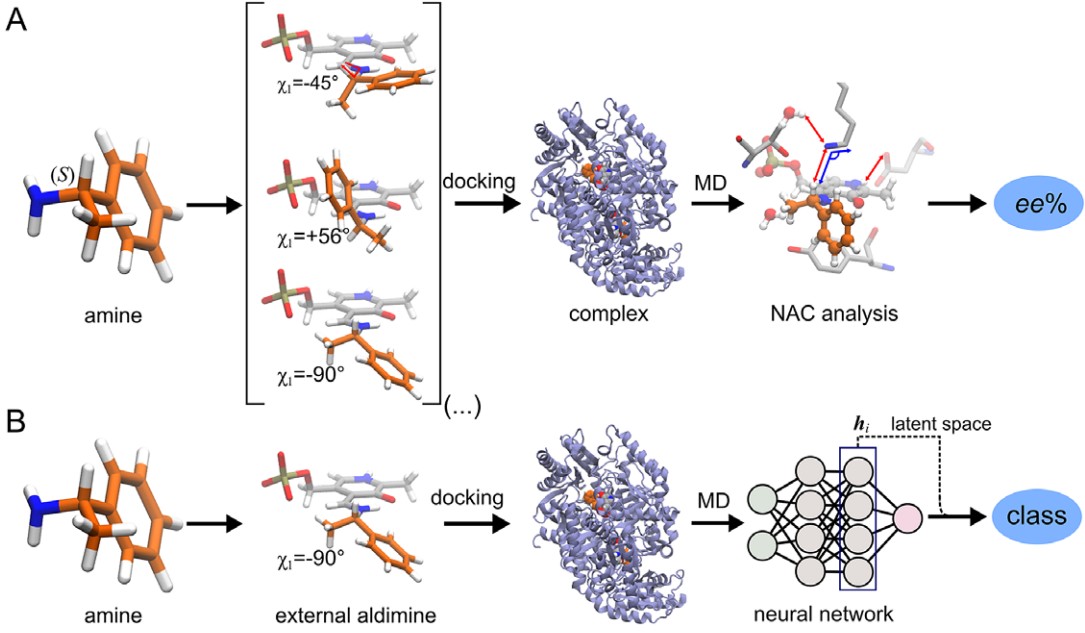

**Fig. 1.** Computational prediction of the ω-TA selectivity by (*a*) geometric analysis of near-attack conformations (NACs) and (*b*) deep learning. In both methodologies, short MD simulations (20 ps) are run using the docked complex as starting conformation. In the NAC-based methodology, presented in Ramírez-Palacios *et al.* (2021), a combination of docking and MD simulations was needed to sample through a slow-moving dihedral ($\chi_1$). The DL-based methodology, presented in this work, does not necessitate such a strategy of sampling through slow-moving parameters, which reduces the number of MD simulations required. Furthermore, no hand-crafted geometric features are needed for the NN to learn to distinguish between trajectories that contain 'reactive' or 'non-reactive' enantiomers, which would reduce the number of man-hours required for its implementation to new systems.

**Fig. 2.** Structures of the compounds used in this study. The ligands were the external aldimine form of the amines shown in the figure. Following CIP rules, all compounds shown are (*S*)-enantiomers, except **32**, **33**, **35**, **36** and **45** which formally are (*R*)-enantiomers.

of geometric criteria used to define binding poses or to identify interesting events taking place in the trajectories that might characterise the classes. Many deep-learning models are explainable by design (intrinsic interpretability), while others are built as black boxes and obtaining explanations requires actions (*post hoc* interpretability; Bodria *et al.,* 2021). Owing to the visual nature of the representations used as input to train CNN models (and the maturity of the architecture), convolutional networks allow a more visual description of the decision-making process in the form of saliency maps. A saliency map shows the relative contribution from each input region (pixel) to the final prediction. One of the first saliency maps proposed was class activation map (CAM; Zhou *et al.,* 2016) but it requires the feature maps to directly precede the softmax layers (which means that not every topology allows this type of saliency map). CAM was later improved to gradient-weighted class activation map (Grad-CAM) and guided Grad-CAM (Selvaraju *et al.,* 2017). Other saliency maps worth mentioning are LIME (Ribeiro *et al.,* 2016), RISE (Petsiuk *et al.,* 2018), IntGrad (Sundararajan *et al.,* 2017) and SmoothGrad (Smilkov *et al.,* 2017). Bodria *et al.* (2021) provide an excellent up-to-date review on explanation methods.

## Methods

### MD simulations

Enzyme–ligand complexes obtained by Rosetta docking were used as starting conformations for 20 ps-long atomistic MD simulations. The enzyme is the wild-type *Vf*-TA (PDB: 4E3Q), and the ligand is the external aldimine intermediate of a set of 49 compounds (Fig. 2, Fig. 3). Exact details about the methodology for

running the MD simulations was presented in Ramírez-Palacios *et al.* (2021).

### Dataset construction

The dataset was constructed from 9,800 MD trajectories (100 per ligand), half belonging to the class labelled as 'reactive' and the other half to the class 'non-reactive'. The trajectories containing the preferred enantiomer [typically the (*S*)-enantiomer] were labelled as 'reactive', whereas the trajectories containing the non-preferred enantiomer [typically the (*R*)-enantiomer] were labelled as 'non-reactive'. The input tensors used for training the DL models consisted of 15 descriptors (distances, angles and dihedrals) taken from each 1,000-frame trajectory (20 ps of simulation), $x_i \in \mathbb{R}^{N \times F}$, where $N$ is the number of timesteps and $F$ is the number of descriptors per trajectory. The descriptors are distances, angles or dihedrals that are considered important for the transamination reaction to take place (Fig. 3, Fig. 4; Cassimjee *et al.,* 2015). Descriptors relating to a water molecule near Lys285 and Thr322 were also included. Some descriptors are redundant (e.g. $d_5$ and $d_6$). The training dataset was split into training (80%) and validation sets (20%). To facilitate – but not guarantee – generalisation to unseen ligands, the split between training and validation sets was not random but ligand-dependent: all trajectories containing ligands **01**–**38** were assigned to the training dataset, and all trajectories containing ligands **39**–**49** were assigned to the validation dataset.

### Class labels

Each trajectory was labelled as either 'reactive' or 'non-reactive', depending exclusively on whether the ligand contained in the complex

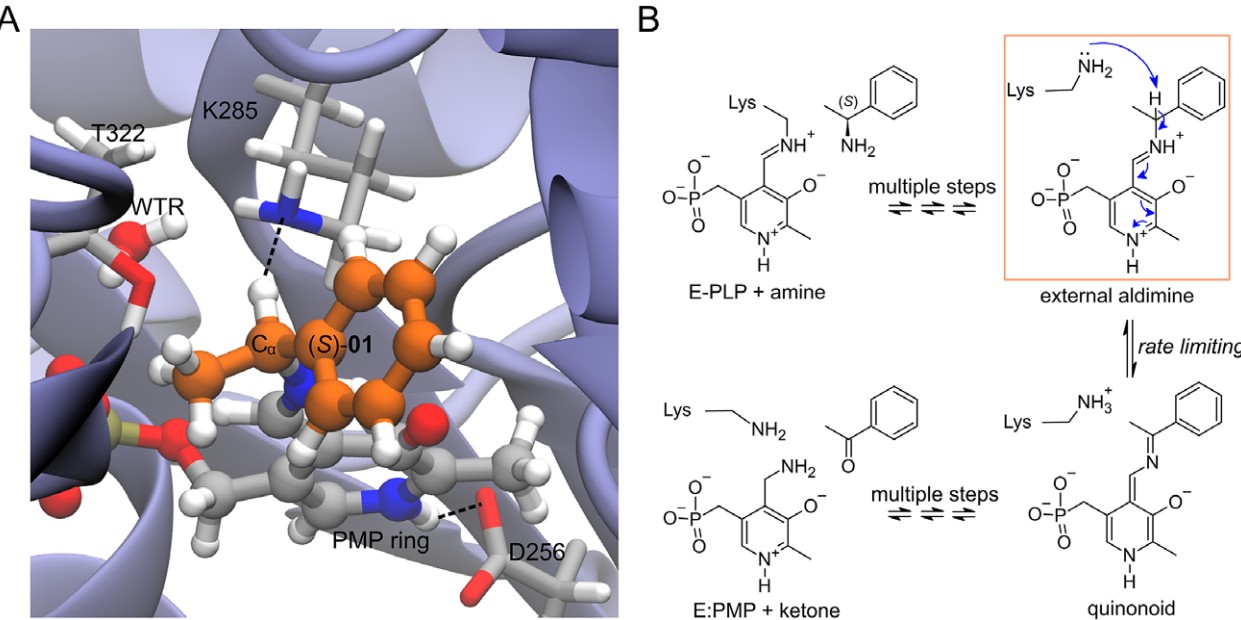

**Fig. 3.** (*a*) Structure of the complex of *Vf*-TA with the external aldimine intermediate of compound (*S*)-**01**. (*b*) Transamination reaction. The fully reversible transamination reaction consists of multiple steps in which an amino group is transferred from the substrate to the cofactor or, in the reverse direction, from the cofactor to the substrate. Among the reaction intermediates, we only modelled the external aldimine intermediate, because it is most likely involved in the rate-limiting step of the reaction. In this step, the nucleophilic proton abstraction of the external aldimine by the catalytic lysine leads to the formation of the quinonoid intermediate. Short MD simulations of the external aldimine intermediate complex were used as input for the neural networks, which were tasked with classifying them.

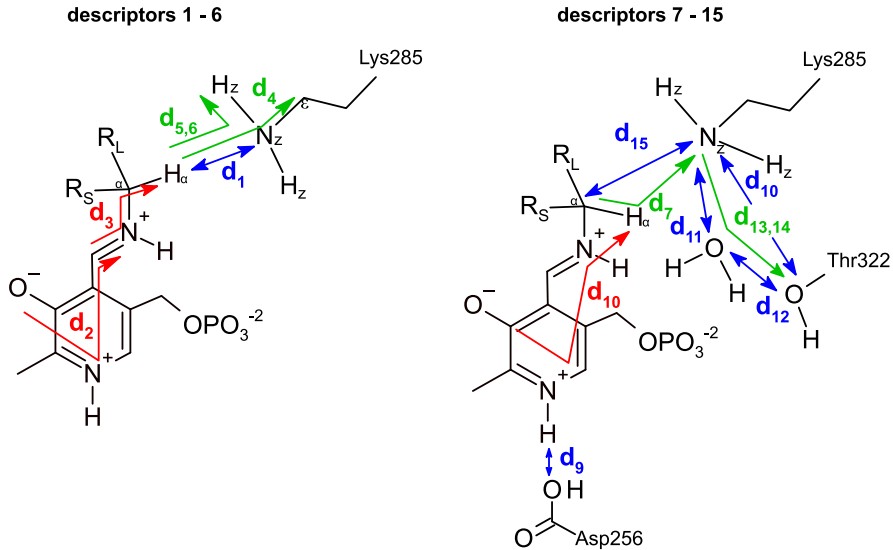

**Fig. 4.** Descriptors used to represent the MD trajectories. Colour codes are: *blue*, distances; *green*, angles; *red*, dihedrals. The 15 descriptors are shown in two parts only for clarity.

was the preferred enantiomer (class 'reactive') or the non-preferred enantiomer (class 'non-reactive'). The label is referring to whether the ligand can be accommodated by *Vf*-TA and lead to catalysis. Typically for *Vf*-TA, the (*S*)-enantiomer is the preferred enantiomer (Fig. 2). The assumption is that all simulations obtained from the preferred enantiomer would behave similar among each other but dissimilar from the simulations of the opposite enantiomer, and vice versa. Here, the word *similar* is in terms of the metric found by the neural network to better accomplish the binary classification task.

## Network building and training procedure

All models were built using the *TensorFlow* 2.4 library (Abadi *et al.*, 2016; sequential API). Unless otherwise indicated, all layers use the rectified linear unit (ReLU) activation function. Hereby a description of the construction and training procedure for the CNN (1D- and 2D-CNN) and LSTM models for the binary classification of MD trajectories, as well as the CAM and latent space visualisation (Fig. 5).

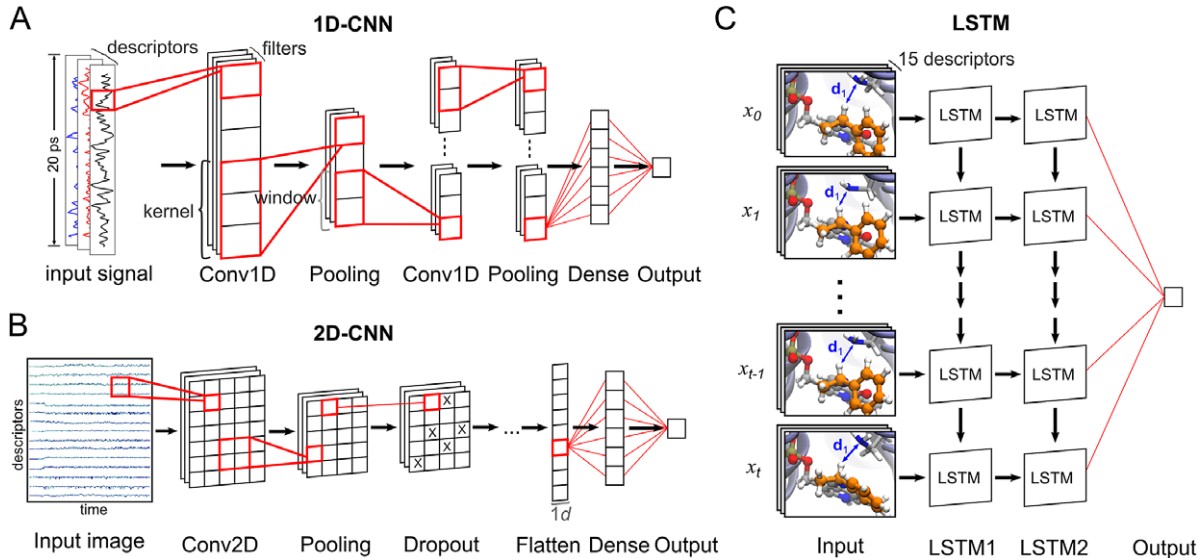

**Fig. 5.** Configuration of the NNs used to classify the MD trajectories. Each trajectory consists of 1,000 frames and each frame is represented by 15 descriptors (distances, angles and dihedrals thought to be relevant in the studied reaction). (*a*) In the 1D-CNN model, an MD trajectory is treated as a signal with multiple channels (descriptors). The input signal (sized 1,000 × 15) passes through multiple convolutional, pooling and dense layers until the final neuron outputs 0 or 1 to represent the class 'non-reactive' or 'reactive', respectively. (*b*) In the 2D-CNN model, the trajectory is represented as an image of size 1,000 × 15 (binary classification) or 320 × 320 (CAM visualisation), with one colour channel (analogous to a black-and-white picture). The input image passes through multiple convolutional, pooling, dropout and dense layers until the final neuron outputs the class prediction. (*c*) The LSTM model is trained to predict the next frame based on information from all other frames that came before it. After training, the embeddings obtained from the second LSTM layer are used to predict the trajectory's class.

### 1D-CNN model

Input vectors for the 1D-CNN model were of shape [batch,1000,15]. The 1D-CNN model was constructed using the following stack of Keras layers: (1) Conv1D (16 filters, kernel of length 7), (2) MaxPooling1D (size of pooling window = 5), (3) Conv1D (16 filters, kernel of length 7), (4) GlobalMaxPooling1D and (5) Dense (one output unit, sigmoid activation). The network was trained via backpropagation for 10 epochs using the RMSprop (learning rate = $1 \times 10^{-4}$) optimiser in minibatches of eight examples. Binary cross-entropy was used as the loss function.

### 2D-CNN model

Input vectors for the 2D-CNN model were of shape [batch, 15,1000,1]. The model was constructed using the following stack of Keras layers: (1) Conv2D (eight filters, kernel of size 1 × 5), (2) MaxPooling2D (pool size = 1 × 3), (3) Dropout (dropout rate = 0.3), (4) Conv2D (four filters, kernel of size 1 × 5), (5) MaxPooling2D (pool size = 1 × 3), (6) Flatten, (7) Dense (one output unit, sigmoid activation). The network was trained in minibatches (size 16) via backpropagation for 20 epochs using the RMSprop (learning rate = $1 \times 10^{-4}$) optimiser. Binary cross-entropy was used as the loss function.

### LSTM model

The LSTM model was constructed using the following stack of Keras layers: (1) LSTM (16 units, tanh activation function), (2) LSTM (8 units, tanh activation function), (3) Dense (1 output unit, without activation). Input vectors were of shape [batch,50,15]. The network was trained for five epochs using the ADAM optimiser (Kingma and Ba, 2017; learning rate = 0.01) with a batch size of 128. The dataset contained ~500,000 input vectors. The task of the LSTM model can be simplified as: given the following input vector containing 15 descriptors measured at 50 timesteps ($x_i \in \mathbb{R}^{50 \times 15}$), predict the value of the query descriptor ($\mathbf{d}_4$) at the next timestep ($\widehat{y}_i \in \mathbb{R}$). The mean-squared error (MSE) between the predicted ($\widehat{y}_i$) and true ($y_i$) label was used as the loss function. After training the LSTM model, the embeddings obtained from the penultimate layer ($h_i^{LSTM} \in \mathbb{R}^8$) were used as input data.

### 2D-CNN model for CAM visualisation

The 2D-CNN model used to create the activation maps used pixel images as input vectors. The input vectors were created using the ImageDataGenerator Keras class to resize the trajectories to [batch,320,320,1]. The overparameterised model was constructed using the following stack of Keras layers: (1) Conv2D (128 filters, kernel of size 3 × 3), (2) MaxPooling2D (pool size = 2 × 2), (3) Conv2D (64 filters, kernel of size 3 × 3), (4) MaxPooling2D (pool size = 2 × 2), (5) Conv2D (32 filters, kernel of size 3 × 3), (6) MaxPooling2D (pool size = 2 × 2), (7) Conv2D (16 filters, kernel of size 3 × 3), (8) MaxPooling2D (pool size = 2 × 2), (9) Flatten, (10) Dense (256 units) and (11) Dense (one output unit, sigmoid activation). The network was trained in minibatches (size 1) via backpropagation for 20 epochs using the RMSprop (learning rate = $1 \times 10^{-4}$) optimiser. Binary cross-entropy was used as the loss function.

### Latent space visualisation

A two-dimensional projection of the vector embeddings produced by the penultimate layer of the 1D-CNN ($h_i \in \mathbb{R}^{16}$) model were produced by t-SNE (t-distributed stochastic network embeddings; van der Maaten and Hinton, 2008), as implemented in the SkLearn library (Pedregosa *et al.*, 2011; with perplexity = 30).

## Results and discussion

### *Dynamics of the studied system*

The length of the MD trajectories was capped at only 20 ps to keep the computational cost low, which is useful for its applicability in high-throughput screening. Longer simulations in the order of

hundreds of ns can easily differentiate between 'reactive' and 'non-reactive' enantiomers, because the 'non-reactive' enantiomer has enough time to evolve and adopt a non-catalytic orientation ($\chi_1 = \mathbf{d}_3 = +90°$), whereas the 'reactive' enantiomer remains in a catalytic orientation ($\chi_1 = \mathbf{d}_3 = -90°$; Ramírez-Palacios et al., 2021). However, the initial conformation is mostly retained during the shorter simulations, as shown in Supplementary Fig. 1. We were unable to tell the two classes apart by visual inspection of the trajectories, and wondered whether a NN could.

### Distribution of input descriptors

To assess whether a NN is even needed to classify the trajectories, one can first look at the distribution of the input descriptors to see if differences between classes are already present. The distribution of descriptors used as input data shows no difference in distribution between the two classes (Supplementary Fig. 2). This means that any algorithm used for classification of the trajectories cannot rely solely on the position-dependent values of one descriptor, and instead needs a combination of descriptors [as was shown in the previous work (Ramírez-Palacios et al., 2021)] or patterns (e.g. time-dependent changes in position). CNNs can do both, and are thus a good choice to classify the trajectories.

### Trained CNNs achieve high accuracy in the binary classification of MD trajectories

The trained CNNs achieved a good accuracy in the binary classification task in both the training and validation datasets (Table 1; see below for the LSTM model). The training and validation datasets contained trajectories from ligands **01–38** and **39–49**, respectively. The split between training and validation datasets was done based on the ligand identity instead of randomly, to allow testing the model's generalisation to unseen ligands. While it is impossible to know for certain whether the probability distribution of the training dataset is large enough to cover every possible ligand, the high accuracy the model achieves in the validation dataset, which contains unseen ligands, does suggest some generalisation. Fig. 6 shows that most ligands in both the training and validation datasets had a low number of misclassified trajectories. For example, ligand **01** had only one misclassified trajectory: the trajectory belonged to the class 'reactive' (*blue*) but the NN incorrectly classified it as belonging to the class 'non-reactive' (*red*). The ligand with the highest number of misclassified trajectories was ligand **35** (see Ramírez-Palacios et al., 2021 for more details on ligands **35** and **36**). Most ligands had only a small number of misclassified trajectories, and 11 of them did not have any misclassified trajectory.

### Disentanglement of input channels

After seeing the 1D-CNN model was accurate in the classification of trajectories, an obvious step forward is to understand what the NN is looking at to arrive to a final decision about the class to which the input vector belongs. The simplest would be to determine on which descriptors (input channels) the NN relies the most. However, disentanglement of the input channels is not an easy task (Cui et al., 2020), because individual channels are not processed individually in a 1D-CNN architecture. For this reason, an ablation study was performed to evaluate the contribution of each channel to the model's performance. An ablation study consists of evaluating the performance of the model after removing one or more of its components (Sheikholeslami et al., 2021). To get an overall picture about the importance of each channel, 15 independent 1D-CNNs were constructed, and each were trained and evaluated using only one descriptor as input vector ($x_i \in \mathbb{R}^{N \times 1}$). While the approach of using one descriptor at a time does not tell the whole story (because it ignores synergies between descriptors), it does give an impression of which descriptors might be important, for example, for the enzymatic reaction to take place. The results of these 15 1D-CNNs are presented in Fig. 7. For the neural network, descriptor $\mathbf{d}_1$ is not a good descriptor by itself to discriminate between the two classes, which is surprising, because $\mathbf{d}_1$ corresponds to the distance from the proton to be abstracted to the attacking atom ($H_\alpha - N_z$), and thus a close distance would be needed for the reaction to take place. And similarly, neither is the distance between $N_z - C_\alpha$ ($\mathbf{d}_{15}$) a good descriptor for the classification task. It is important to clarify that this result is only telling us that neither $\mathbf{d}_1$ nor $\mathbf{d}_{15}$ alone are enough for the NN to do the classification task, but a combination with other descriptors might produce a different outcome. Fig. 7 shows that while the distance from the water molecule to the oxygen in Thr-OH ($\mathbf{d}_{12}$) is not important, the distance from the same water to the Lys-NH2 is important ($\mathbf{d}_{11}$). Lys285 is believed to require water assistance for proton abstraction from the external aldimine intermediate, but there is an alternative mechanism that does not require water (Cassimjee et al., 2015). The descriptor that better achieves a good accuracy is the angle between $H_\alpha - N_z - C_E$ ($\mathbf{d}_4$), which is expected, because Lys285-NH$_2$ needs to be positioned at the correct angle for the nucleophilic proton abstraction to take place. An interesting observation is that the two 'siblings' of $\mathbf{d}_4$ [$\mathbf{d}_5$ and $\mathbf{d}_6$, both refer to the same angle ($H_\alpha - N_z - H_z$; see Fig. 4)] do not achieve the same accuracy as $\mathbf{d}_4$. The observation is interesting, because in the previous protocol (Ramírez-Palacios et al., 2021), the angles between $H_\alpha - N_z - H_{z1,2}$ ($\mathbf{d}_{5,6}$, previously known as $\theta_1$ and $\theta_2$) were part of the geometric criteria used to define the reactive conformations of interest, but these results suggest that using the angle $H_\alpha - N_z - C_E$ instead might have been a better choice.

**Table 1.** Results of the tested NNs in the binary classification of MD trajectories

| Model | Training | | | Validation | | |
|---|---|---|---|---|---|---|
| | ↑Acc. | ↓Loss | ↑ROC-AUC | ↑Acc. | ↓Loss | ↑ROC-AUC |
| 1D-CNN | 0.93 | 0.24 | 0.99 | 0.91 | 0.25 | 0.93 |
| 2D-CNN | 0.91 | 0.22 | 0.97 | 0.89 | 0.24 | 0.91 |
| LSTM | 0.50 | 0.69 | 0.50 | 0.50 | 0.70 | 0.50 |

Abbreviations: LSTM, long short-term memory; ROC-AUC, area under the curve of the receiver-operator characteristic curve.

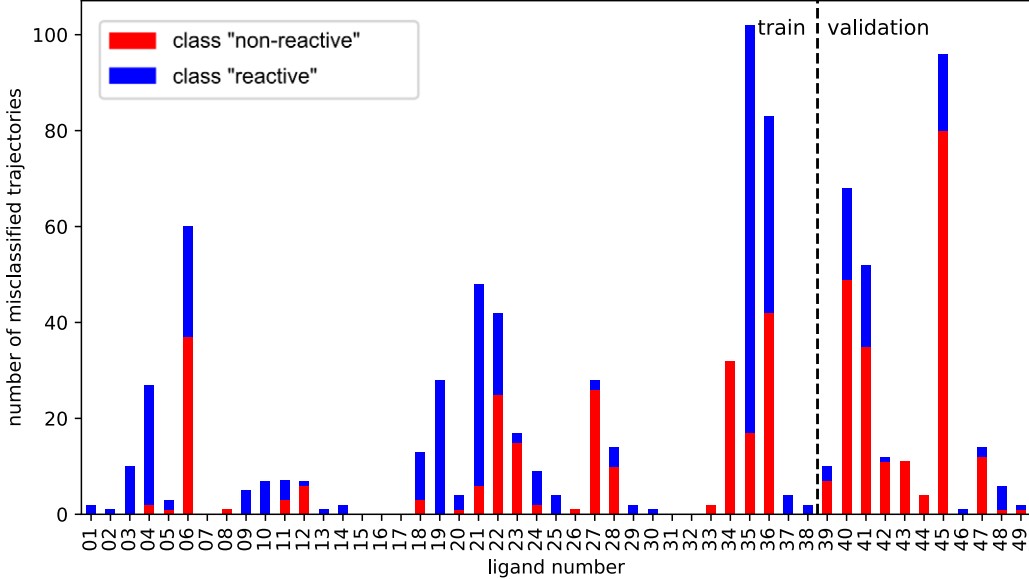

**Fig. 6.** Bar plot showing the number of misclassified trajectories per ligand molecule when evaluated with the trained 1D-CNN model. We want the number of misclassified trajectories to be small. The total number of trajectories per ligand is 200, half of which are labelled as 'non-reactive' and the other half are labelled as 'reactive'. For example, ligand **06** has 100 trajectories containing the non-preferred enantiomer (class 'non-reactive') of which 37 were misclassified (red bars), and 100 trajectories containing the enantiomer preferred by *Vf*-TA (class 'reactive') of which 23 were misclassified (blue bars) by the NN. There are 49 ligands in total, adding up to a total of 9,800 trajectories.

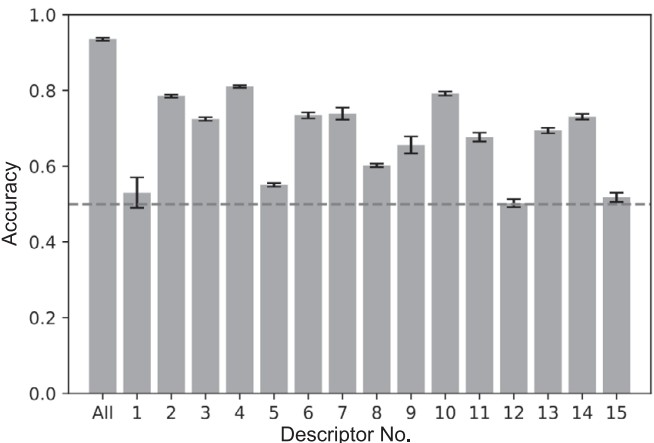

**Fig. 7.** Accuracy obtained by using one input channel for training, $x_i \in \mathbb{R}^{N \times 1}$, and the accuracy obtained by using all the input channels, $x_i \in \mathbb{R}^{N \times 15}$ (first bar, labelled as 'All'). The error bars are standard deviations over five replicas.

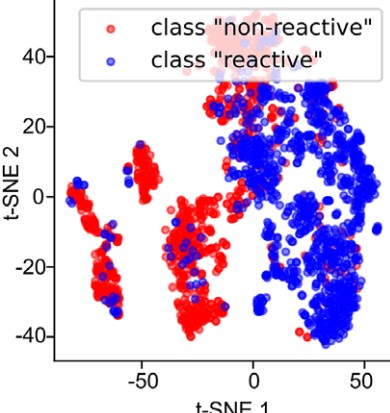

**Fig. 8.** t-SNE of the latent space representations ( $h_i \in \mathbb{R}^{16}$ ) obtained from the penultimate layer of the 1D-CNN model. The evaluations were made in the validation dataset ($n = 2,200$ examples). The reader can think of the latent space as follows: imagine you show an image of a molecule to a chemist and scan their neurons to see which neurons are active and which are not. You then repeat the experiment with a non-expert in chemistry and compare the two scans. The scans will be different, because the expert can give meaning to the image shown. We expect the scan of the neuron activations of the expert to tell us something about the molecule that is not necessarily encoded in the input image (e.g. whether the molecule is realistic or not), coming from the experience (training) of the expert (neural network). That is latent space.

## Visualisation of the latent space representations

Visualising the latent space representations ( $h_i$ ) onto which the trained models project the input vectors ( $x_i \; h_i, x_i \in \mathbb{R}^{1000 \times 15}$, $h_i \in \mathbb{R}^{16}$) is useful, because it can provide visual clues about whether subclasses are present (Fig. 8). The presence of subclasses would indicate that on top of trajectories being *different*, because they belong to distinct classes, they are also different in some other way. Here, the word different is according to the metric the NN learned through training as best-suited to do the classification task. The latent space visualisation did not reveal any major subclasses for the class 'reactive', but was indicative of 3–4 subclasses (i.e. clusters of points separated from each other) for the class 'non-reactive' (Fig. 8). However, because the LSTM approach was unsuccessful in the binary classification task (see later), these subclasses were not investigated any further.

## CAM visualisation

Visualising how a model arrives to a particular decision about the input data can by itself be very useful, sometimes more than the task the model was assigned to perform (see *below*). Fig. 9 shows the CAM visualisation from the 1D-CNN model. The activations are evenly spread out throughout the trajectory which indicates that the neural network is looking at processes happening throughout the trajectory and not at just some specific portion of it. It also suggests that shorter trajectories (<<20 ps) would be enough to do the

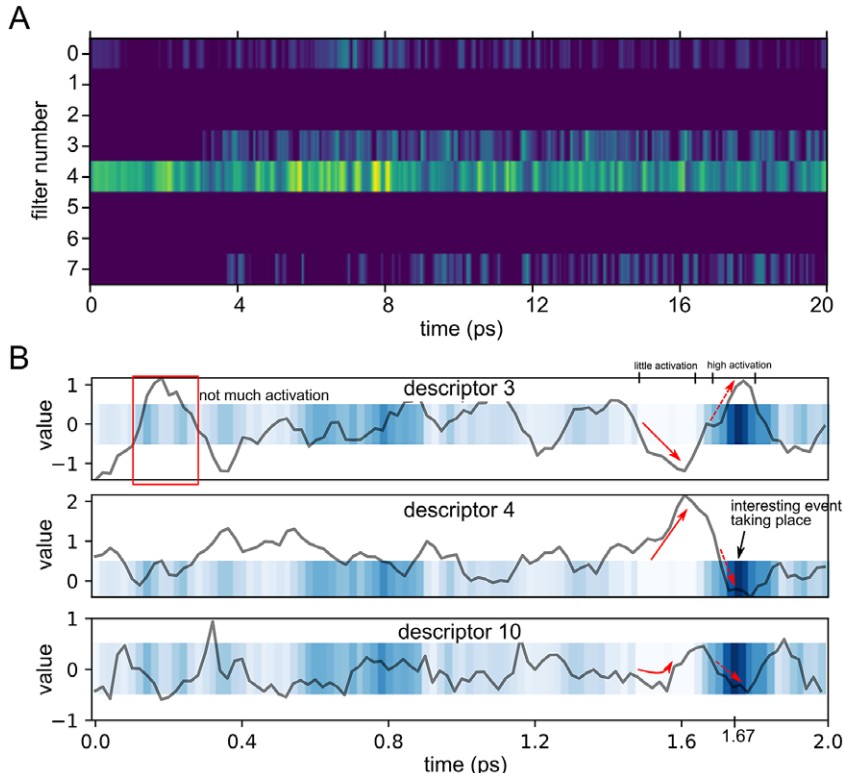

**Fig. 9.** (*a*) Heatmap showing the CAM activations of the 1D-CNN model of one input trajectory, presented as example. The *y*-axis is the filter number, and the *x*-axis is the time dimension. (*b*) CAM activations (heatmap) of filter number 4 overlapped to the input vectors of descriptors 3, 4 and 10 (line plot), along a small 2.0 ps window (*x*-axis). The *y*-axis contains the normalised values from each descriptor. The hand-drawn arrows show the direction each descriptor follows. At 1.67 ps, the activations are at their highest, and at 1.55 ps, activations are low. The red rectangle shows a region with little activations but where descriptor 3 looks visually similar to the region in which the activations were at their maximum (~1.67 ps).

classification (Supplementary Fig. 3). An overlap of the input descriptors and the activations produced (*Fig. 9b*) shows the dependency of the activations on the input signal. Some activations occur when the descriptor reaches a peak, a dip or a plateau, but the activation comes from all the descriptors taken together as a whole. This type of visualisation can be useful in identifying the temporal location of interesting molecular events.

Visualisation of 2D-CNN is more illustrative than that of 1D-CNN. For this reason, a second CNN model was trained, this time using a 2D-CNN architecture. The CAM visualisation is presented in Fig. 10. As it can be seen, the neural network activates (*yellow*) in the regions of the 2D image where a trajectory signal is present. As expected, activations were low in the regions in-between channels.

As mentioned earlier, disentanglement of the contributions of each input descriptor to the classification is not an easy task. A way to make use of the trained model is to visualise the position-dependent activations of each descriptor. This would be useful, for example, in helping define reactive poses or simply to know which values are preferred. For example, in Supplementary Fig. 2 it is shown that descriptor $\mathbf{d}_1$ has the same distribution in the two classes, which means that one cannot simply choose a cut-off value for $\mathbf{d}_1$ and hard-code a solution for the classification task. Fig. 11 shows that the 1D-CNN model has no apparent position-dependent preference for descriptors $\mathbf{d}_1$, $\mathbf{d}_8$, $\mathbf{d}_9$, $\mathbf{d}_{12}$ and $\mathbf{d}_{15}$, as both classes show the same distribution. On the other hand, descriptor $\mathbf{d}_4$, the angle between $H_\alpha - N_z - C_E$, is preferred to be large for the NN to classify the simulation as 'reactive' and small to classify it as 'non-reactive'. This makes sense as a larger angle implies that the

electrons in $N_z$ can be pointing directly towards $H_\alpha$ (Fig. 4). Descriptor $\mathbf{d}_{11}$ is similar: the NN prefers it to have small values for 'non-reactive' trajectories and large values for 'reactive' trajectories. Descriptor $\mathbf{d}_{11}$ is the distance between Lys285 and a water molecule in the vicinity of Lys285 and Thr322. Nevertheless, it must be noted that everything rationalised in this paragraph about Fig. 11 is only true for the filter number 4 of the trained 1D-CNN model, and that other filters might have different criteria for discerning between classes. Furthermore, Fig. 11 shows only the position-dependent component that makes the activation happen, and does not include the pattern-dependent component or the individual effect that each descriptor had in the activation. Therefore, while Fig. 11 can be useful, conclusions should be drawn with care of the context.

### Semi-supervised learning through LSTM

This approach was not successful in either the semi-supervised learning or the binary classification tasks. First, the results of the LSTM model on predicting the values of descriptor $\mathbf{d}_4$ in the 51st timestep given the values of all descriptors ($\mathbf{d}_1$–$\mathbf{d}_{15}$) in the previous 50 timesteps gave a MSE of 0.114. For reference, the results of simpler models are as follows: baseline (0.155), linear (0.154), dense (0.152), 1D-CNN (0.121); where the number in parenthesis is the accuracy from the validation dataset (Supplementary Fig. 4). We tried, unsuccessfully, to improve the performance of the LSTM model by using time-lagged input data (i.e. instead of using every frame, use every third or fifth frame; Zeng *et al.*, 2021) but the performance of the LSTM model was still poor. Therefore, it is not

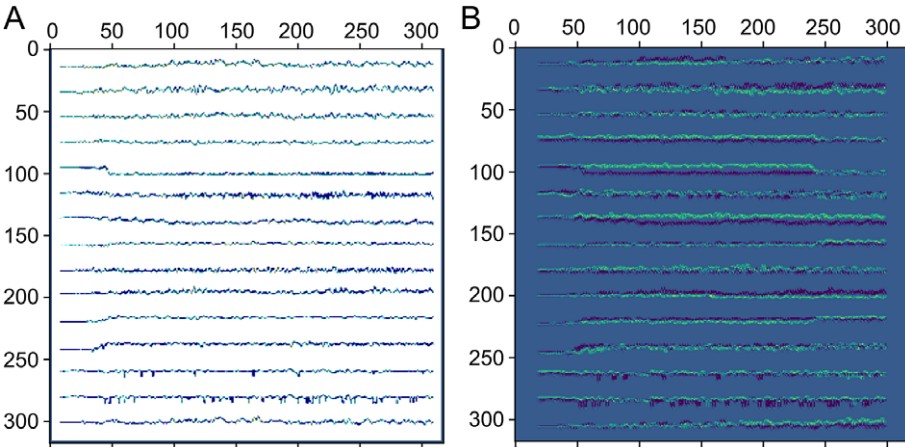

**Fig. 10.** (a) Example of an input image of size 320 × 320 pixels, and (b) the CAM produced by the image after passing it through a 2D-CNN. Viridis colourmap: the regions with the lowest activations (low importance) are coloured *blue*, the regions that produce the highest activations are in *yellow* and the regions with middle importance are in *green.* The highest activations come from the pixels containing trajectory information.

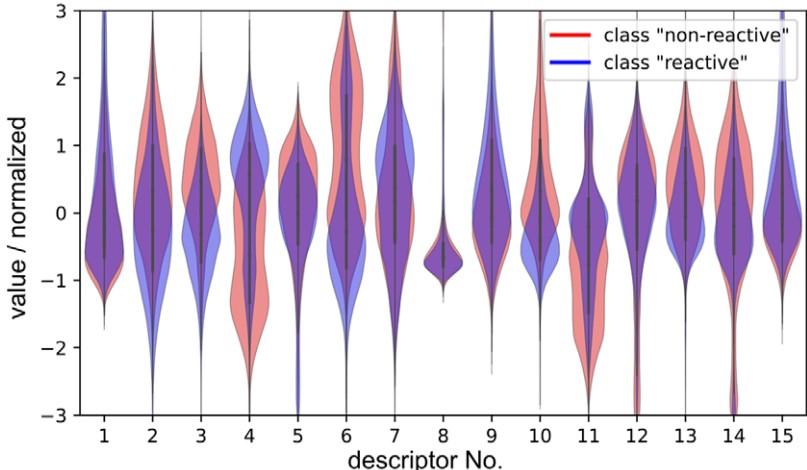

**Fig. 11.** Violin plots showing the position-dependent values (*y*-axis) at which each descriptor (*x*-axis) showed the maximum CAM activation to be classified as either class 'reactive' (*blue*) or class 'non-reactive' (*red*). The plot shows all 9,800 trajectories. CAM visualisation was obtained from the 1D-CNN model (filter 4 of Fig. 9).

surprising that using the embeddings generated from the LSTM model from the time-series ( $x_i$ $h_i^{LSTM}, x_i \in \mathbb{R}^{1000 \times 15}, h_i^{LSTM} \in \mathbb{R}^8$ ) as input data for binary classification was unsuccessful in separating the time-series as belonging to either of the two classes (loss = 0.699, acc = 0.500).

### *Results suggest an alternative to the sampling strategy presented in our previous work*

We had presented a protocol for predicting the enantioselectivity of *Vf*-TA towards a query compound (Ramírez-Palacios *et al.,* 2021). The protocol consisted in counting the number of frames in which a set of geometric criteria were simultaneously met. Because one of the descriptors ($\mathbf{d}_3$, known as $\chi_1$ in the previous work) evolved too slowly for a 20 ps simulation to sample enough conformational space, a combination of docking and MD simulations was needed. The method hereby presented does not necessitate such a strategy (Fig. 1). Instead, all complexes are docked in a *catalytic orientation* ($\chi_1 = -90°$) and a neural network analyses the simulations. Even with the slowly moving descriptor, $\mathbf{d}_3$, the neural network is still

able to extract patterns that differentiate trajectories from the class 'non-reactive' and the class 'reactive' (Fig. 7: descriptor 3). Such patterns are difficult to describe but one can imagine, for example, that the descriptor of one class might move up–down–up–down, whereas the descriptor of the other class might move up–up–down–down. Finding what those patterns are could also help refine the geometric criteria formerly used to predict the *Vf*-TA enantioselectivity. For example, our current analysis points to the importance of the angle $H_\alpha - N_z - C_E$, which could be included in the NAC definition to obtain better predictability.

### Conclusions

Using DL to analyse MD simulations leverages the adaptability of neural networks to learn without manual intervention which descriptors are important in telling classes apart. The trained CNNs were capable of binary classification of MD trajectories with high accuracy, but the same is not true for the LSTM model. Knowing which descriptors are important can be useful in searching and fine-tuning geometric parameters to be used in the definition of binding

poses. Furthermore, the trained models can highlight the time regions of the trajectory where interesting events (i.e. events that are important for discerning classes) are taking place. An updated framework is still needed to better explain the manner in which the neural network arrives to its final decisions. Better explanations could facilitate the integration of DL into MD workflows.

## Abbreviations

CAM class-activation map
CNN convolutional neural network
DL deep learning
LSTM long short-term memory
MD molecular dynamics
NN neural network
RNN recurrent neural network
*Vf*-TA ω-transaminase from *V. fluvialis*
ω-TA ω-transaminase

**Open peer review.** To view the open peer review materials for this article, please visit http://doi.org/10.1017/qrd.2022.22.

**Supplementary Materials.** To view supplementary material for this article, please visit http://doi.org/10.1017/qrd.2022.22.

**Acknowledgements.** C.R.-P. thanks CONACYT for the doctoral fellowship.

**Conflicts of interest.** The authors declare no conflicts of interest.

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
