## [Reviewer Report]

*Comments to Author*: The paper reports the use of sophisticated deep learning methods to analyse molecular dynamics trajectories. Such approaches have great promise for the future, so this paper is certainly of interest to the molecular dynamics (MD) community. To have the maximum impact, the authors should try and rewrite it so that it is more accessible to the molecular dynamics field, as currently it is very brief when describing the methods used and uses a lot of technical language and jargon that the MD community is currently unfamiliar with (e.g. latent space/input channels).

To improve readability within the MD field, I have the following suggestions:

1) Provide more background to the system of interest and why these calculations are useful. The role and importance of the protein itself has not been discussed.

2) The dataset used has been well described, but how does it interface with TensorFlow? The section on network building in TensorFlow is very technical for an MD audience, and should be described with simpler language.

3) The conclusions should relate more specifically to the chemistry results that were obtained, and the insight into the molecular structures that resulted. Currently it is focused on the technicalities of the machine learning. The attempt to track down the mechanism of the learning and decision making is very interesting, and it would be very helpful if the authors could pull out specific chemical examples from their study to explain what is happening from a structural perspective.

---

## [Reviewer Report]

*Comments to Author*: I believe the manuscript reports on substantial work well supported by the factual information.The results are interesting and important.However, I would prefer to see a more clear connection to commonly understood physical and chemical processes.For example, as simple as drawing the ligand in the active site would help understanding the details of the chemical process.Also, more clear description of the dynamics of the analysed process will help.If I understand correctly, only very short trajectories were analysed (20ps).If that’s the case, what information is contained in such short trajectories?

I think the manuscript can be published after the above is provided.

---

## [Reviewer Report]

*Comments to Author*: Reviewer #1: I believe the manuscript reports on substantial work well supported by the factual information.The results are interesting and important.However, I would prefer to see a more clear connection to commonly understood physical and chemical processes.For example, as simple as drawing the ligand in the active site would help understanding the details of the chemical process.Also, more clear description of the dynamics of the analysed process will help.If I understand correctly, only very short trajectories were analysed (20ps).If that’s the case, what information is contained in such short trajectories?

I think the manuscript can be published after the above is provided.

Reviewer #2: The paper reports the use of sophisticated deep learning methods to analyse molecular dynamics trajectories. Such approaches have great promise for the future, so this paper is certainly of interest to the molecular dynamics (MD) community. To have the maximum impact, the authors should try and rewrite it so that it is more accessible to the molecular dynamics field, as currently it is very brief when describing the methods used and uses a lot of technical language and jargon that the MD community is currently unfamiliar with (e.g. latent space/input channels).

To improve readability within the MD field, I have the following suggestions:

1) Provide more background to the system of interest and why these calculations are useful. The role and importance of the protein itself has not been discussed.

2) The dataset used has been well described, but how does it interface with TensorFlow? The section on network building in TensorFlow is very technical for an MD audience, and should be described with simpler language.

3) The conclusions should relate more specifically to the chemistry results that were obtained, and the insight into the molecular structures that resulted. Currently it is focused on the technicalities of the machine learning. The attempt to track down the mechanism of the learning and decision making is very interesting, and it would be very helpful if the authors could pull out specific chemical examples from their study to explain what is happening from a structural perspective.

---

## [Reviewer Report]

*Comments to Author*: The authors have significantly improved the accessibility and readability of their manuscript, and I am confident it will be of great interest to the biomolecular simulation community.